# Metabolic disorders and post-acute hospitalization in black/mixed-race patients with long COVID in Brazil: A cross-sectional analysis

Ana Paula Andrade Barreto[1,2☯], Marcio Andrade Barreto Filho[1,2☯], Lucimeire Cardoso Duarte[1,2☯], Thiago Cerqueira-Silva[3,4], Aquiles Camelier[1,5], Natalia Machado Tavares[3], Manoel Barral-Netto[3,4], Viviane Boaventura[3,4‡]*, Marcelo Chalhoub Coelho Lima[1,6‡], on behalf of the CPC study group[¶]

**1** Hospital Especializado Octávio Mangabeira (HEOM), Salvador, Brazil, **2** Escola Bahiana de Medicina e Saúde Pública (EBMSP), Salvador, Brazil, **3** Instituto Gonçalo Moniz (Fiocruz-Bahia), Salvador, Brazil, **4** Universidade Federal da Bahia (UFBA), Salvador, Brazil, **5** Fundação Maria Emília Freire de Carvalho (FME), Salvador, Brazil, **6** Universidade Salvador, (UNIFACS), Salvador, Brazil

☯ These authors contributed equally to this work.
‡ VB and MCCL also contributed equally to this work.
¶ Membership of the CPC study group is provided in the Acknowledgments.
* viviane.boaventura@fiocruz.br

**Data Availability Statement:** All relevant data are within the paper and its Supporting Information files.

## Abstract

### Background

Although low-middle income countries have been disproportionately affected by the COVID-19 pandemic, there is scarce information about the impact of long COVID on their population. This study aimed to evaluate long COVID symptomatology, complications (hospital readmission and metabolic disorders), and main clinical features that impact Quality of Life (QoL).

### Methods

This cross-sectional study provides a detailed clinical and laboratory picture of individuals who presented residual symptoms after mild to severe acute COVID-19. Between Aug-2020 to Sep-2021, long COVID patients were evaluated in a reference center for long COVID in Bahia State, Brazil. The EQ-5D-5L questionnaire accessed QoL.

### Results

A total of 1164 (52 ±13.4 years, 57% female, 88% black/mixed-race) were evaluated 2.3 [IQR = 1.6–3.7] months after mild (n = 351, 30.2%), moderate (338, 29.0%) or severe (475, 40.8%) acute illness. Dyspnea (790, 67.9%), fatigue (738, 63.5%), and chest pain (525, 42.9%) were the most frequent residual symptoms regardless of acute severity, affecting the QoL of 88.9% of patients (n/N—826/925), mainly the domains of anxiety/depression and pain/discomfort. High levels of HbA1c were detected for 175 out of 664 patients (26.6%),

**Funding:** MABF received a Scientific Initiation Scholarship - PIBIC from "Conselho Nacional de Desenvolvimento Científico e Tecnológico" (CNPq) -Brazil; T.C.-S. is a PhD student at the Post-Graduation Program in Health Sciences-UFBA, which is supported by the "Coordenação de Aperfeiçoamento de Pessoal de Nível Superior"-Brasil, finance code 001. A C. reports consulting fees from "Fundação Maria Emilia Freire". MB-N and VSB are research fellows from CNPq. This project was partially supported by FAPESB - "Fundação de Amparo à Pesquisa do Estado da Bahia" (grant no. SUS0033/2021 - PPSUS 02/ 2020). The funders had no role in study design, data collection and analysis, decision to publish, or preparation of the manuscript.

**Competing interests:** The authors have declared that no competing interests exist.

40% of them without a previous diagnosis of diabetes mellitus. Of note, hospital admission one-to-three months after the acute phase of disease was required for 51 (4.4%) patients.

## Conclusion

In this majority-black/mixed-race population, long COVID was associated with post-acute hospitalization, newly diagnosed diabetes mellitus, and decreased QoL, particularly in women and regardless of disease severity of acute infection, suggesting important implications for health care system.

## Introduction

Almost two years after World Health Organization declared COVID-19 a pandemic, more than 345 million people have been infected, many of whom persist with symptoms [1, 2]. This clinical condition, recently named long COVID [3], involves early and late-onset symptoms, including respiratory, musculoskeletal, and sensorineural complaints detected no less than four weeks after the disease onset [2].

Most reports on the effect of long COVID have focused on European and North American populations [4–6]. Long COVID has been mainly reported after hospitalization [7, 8], aggravated by post-intensive care syndrome and post-traumatic stress disorders. However, persistent symptoms also affect approximately one-third of patients who experience mild COVID-19 and have been reported to persist for at least six months after the disease onset [4, 9].

COVID-19 disproportionately affects the most vulnerable communities [10, 11]. Brazil is among the most severely affected countries, with almost 29 million confirmed COVID-19 cases as of January 2022 [1]. It is important to note that long COVID has been under-investigated in lower-middle-income countries [12, 13].

Here, we sought to characterize long COVID in Brazilian patients, most of mixed-race and with mild-to-severe disease. We aimed to describe persistent and late-onset symptoms associated with long COVID and complications temporarily related, such as hospital readmission and metabolic disorders. We also evaluated the quality of life (QoL) of patients presenting Long COVID symptoms.

## Methods

### Study design, population, and ethical aspects

We report a cross-sectional analysis of baseline data from a cohort of long COVID cases seen at the Centro Pós-COVID-19 (CPC), a public health outpatient clinic operating at the Octávio Mangabeira Specialized Hospital (HEOM), located in Salvador-Bahia, Brazil. CPC provides outpatient services to residents of the state of Bahia (15.13 million inhabitants). Patients were recruited by invitation upon discharge from local COVID-19 reference hospitals or referred to CPC from outpatient clinics (advertisements were circulated via conventional and social media). The inclusion criteria were: 1) Recruitment at least one month after disease onset; 2) positive SARS-CoV-2 reverse transcriptase–polymerase chain reaction (RT-PCR) test, or IgM/IgG serological test positivity, or a computed tomographic (CT) lung scan compatible with clinical features of viral pneumonia plus epidemiologic criteria. Exclusion criteria: age <18 years, cognitive disorders, pregnancy, or a lack of residual symptoms. The study was approved by the institutional review boards of the Bahia State University (UNEB; protocol no.

38281720.2.0000.0057) and the Santo Antonio Hospital (OSID; protocol no. 33366030.5.0000.0047). All patients provided written informed consent before participating in the study, and the signed consent form was retained in the patients' research chart to document the consent process.

## Clinical and laboratory evaluations

Upon recruitment at CPC, patients were routinely evaluated by a multidisciplinary team (nurse, physician, physiotherapist, nutritionist, psychologist, and social worker). Patients were classified according to reported disease severity during the acute phase of infection (within the first 30 days following disease onset): mild (not requiring hospitalization), moderate (non-ICU hospitalization), or severe (ICU admission).

Patients were recruited by invitation upon discharge from local COVID-19 reference hospitals or came to CPC as outpatients (advertisements were circulated via conventional and social media). All inpatients were referred, but only those with persistent symptoms with outpatient COVID were self-referred/referred in. Data were objectively collected by trained physicians and nurses using a standardized form. For data collection and management, a structured survey created via Research Electronic Data Capture (REDCap) software, with hosting provided by the Gonçalo Moniz Institute (IGM-FIOCRUZ; Bahia, Brazil), was used [14, 15]. Sociodemographic and clinical characteristics were recorded, including respiratory (dyspnea, cough), neurological (headache, gustatory and olfactory dysfunction, dizziness, memory loss), pain (chest pain, myalgia), and constitutional (fatigue, insomnia) symptoms, as well as anthropometric parameters, oxygen saturation, comorbidities, social habits (smoking habit, alcohol use, physical exercise frequency), current/past treatments (e.g., corticosteroids). Dyspnea-associated functional disability was assessed using the Modified Medical Research Council (mMRC) scale, previously validated in Portuguese [16]. A subgroup consisted of patients who provided information on the presence/absence of symptoms during the acute phase of infection and reported the persistence of symptoms or the onset of new symptoms post-acute phase. QoL was assessed using the Portuguese version of the EuroQol (EQ-5D-5L) instrument, which evaluates mobility, self-care, usual activities, pain, anxiety/depression, and general health status [17]. Blood samples were collected to assess the following laboratory parameters: complete blood count, glycemia, hemoglobin A1C (HbA1c), urea, creatinine, AST, ALT, bilirubin, C-reactive protein, sodium, potassium, and creatine kinase. Patients with long COVID who reported hospital admission one month after the onset of disease symptoms were classified as having post-acute hospitalization. The cause and length of post-acute hospitalization admission were obtained through discharge reports and medical inquiries.

## Statistical analysis

All data were analyzed using the R statistical software package [18]. Continuous variables with normal distributions were described as means and standard deviation, whereas variables with non-normal distributions were described as medians and interquartile ranges. Categorical variables were expressed as frequencies and percentages. Data distribution was assessed using the Shapiro-Wilk test and histogram analysis. Missing data were present in some variables and were assumed to be missing completely at random. Data were not imputed, and missing information was described as n/N for each variable. Comparisons of quantitative variables between two independent groups were made using the Student's T or Mann-Whitney tests; for three or more groups, we used ANOVA or Kruskal-Wallis tests. The Chi-square test was used to assess differences between categorical variables. All bivariate analyses were realized separately in each group of disease severity to prevent bias due to different recruitment strategies. The

Benjamini–Hochberg procedure was used to adjust P-value and determine which results were significant. Ordinal logistic regression analysis was performed for each domain of the Euro-Qol, considering age, sex, BMI, hospitalization during the acute phase, presence of medical comorbidities (systemic hypertension, chronic cardiac disease, obesity, asthma, diabetes, chronic obstructive pulmonary disease, chronic kidney disease, and cancer) and the three most prevalent persistent symptoms. For global EuroQol Scores, we performed linear regression. The Brant test was used to test proportionality [19]. Missing data were excluded in regression analysis.

## Results

From August 2020 to September 2021, 1,327 patients with a history of confirmed or suspected COVID-19 were seen at CPC. Patients were excluded if no residual symptoms were presented/reported (n = 81), they refused to participate (n = 58), or were considered ineligible due to pregnancy or severe psychiatric illness (n = 8) (Fig 1). The remaining 1,164 individuals (mean age: 52±13.4 years, 57% female) were classified as having mild (n = 351; 30.2%), moderate (n = 338; 29.0%) or severe (n = 475; 40.8%) disease during the acute phase of infection. Most patients self-reported mixed- or black race (50.9% and 36.7%, respectively) and presented at least one comorbidity (72.6%), mainly hypertension (43.8%) or obesity (43.7%) at the time of their first appointment. Of the patients who were not obese (656), 396 (60.4%) reported weight loss occurring during the acute phase of infection (Table 1).

The median time from symptom onset of COVID-19 to first medical evaluation at CPC was 2.3 months (IQR = 1.6 to 3.7). Dyspnea was the most frequent symptom (789 (67.8%)) associated with long COVID. In addition, 363 (46.0%) presented significant disability,

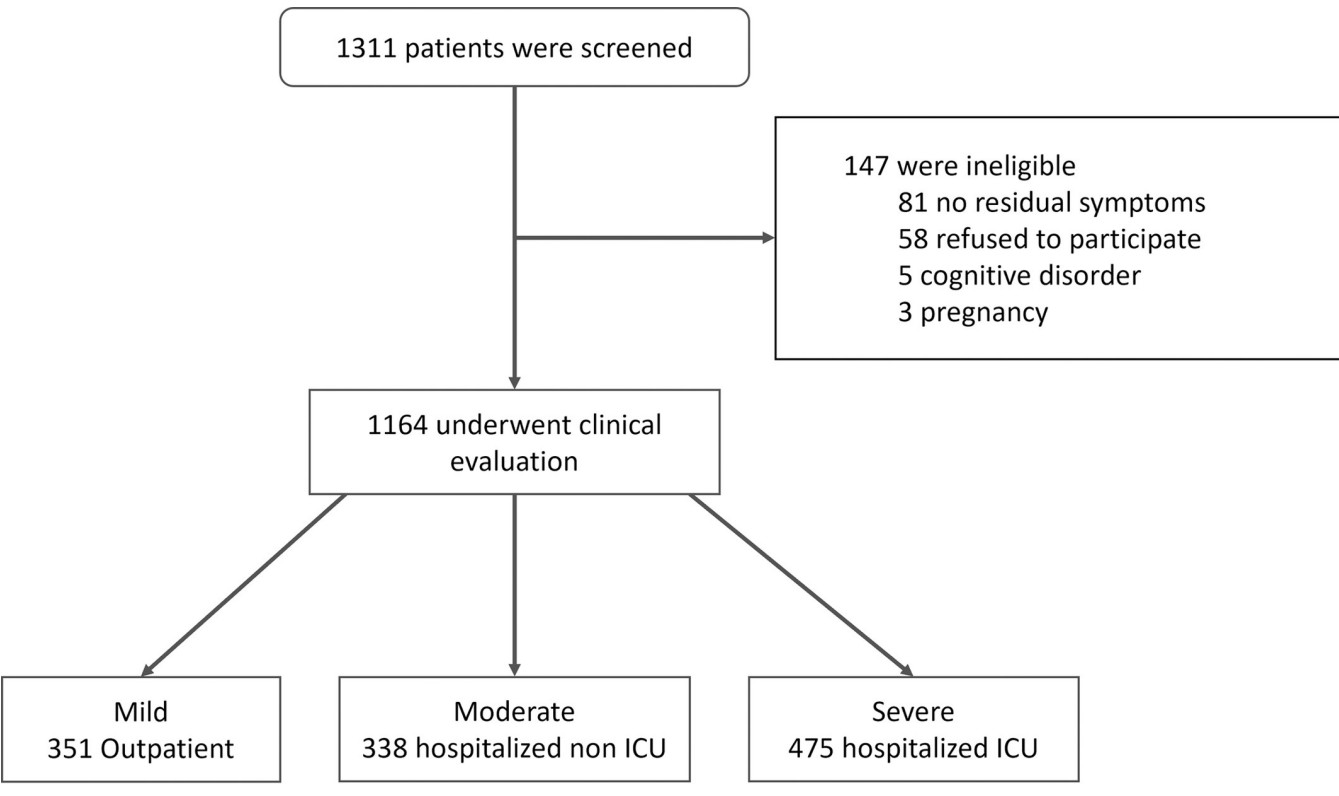

**Fig 1. Flowchart of patients with long COVID seen at CPC (Bahia-Brazil).**

**Table 1. Demographic and clinical characteristics of long COVID patients.**

|  | (N = 1.164) |
|---|---|
| **Age, years** | 52.1 (±13.4) |
| **Female** | 659 (56.6%) |
| **Ethnicity** * | (N = 1162) |
| Mixed-race | 591 (50.9%) |
| Black | 427 (36.7%) |
| White, other | 144 (12.4%) |
| **BMI-kg/m$^2$** | 29.9 (±6.1) |
| **BMI class** |  |
| <30 | 653 (56.3%) |
| 30–34 | 311 (26.8%) |
| 35–39 | 132 (11.4%) |
| >40 | 63 (5.4%) |
| **Time elapsed since disease onset (months)** | 2.3 (1.6–3.7) |
| **SARS-CoV-2 diagnosis**\* |  |
| RT-PCR | 897 (77.0%) |
| Serological (IgM/IgG) | 41 (3.5%) |
| Point of care—Serological | 126 (10.8%) |
| Point of care—Antigen | 108 (9.3%) |
| Radiological + Clinical symptoms | 16 (1.4%) |
| **Comorbidities (≥1)** | 846 (72.6) |
| Hypertension | 510 (43.8%) |
| Obesity | 506/1159 (43.7%) |
| Diabetes Mellitus | 263 (22.6%) |
| Asthma | 103/1161 (8.9%) |
| Cardiovascular disease | 74/1161 (6.4%) |
| Chronic Obstructive Pulmonary Disorder | 34/1153 (2.9%) |
| Other** | 169/1161 (14.6%) |
| **Smoking habit (current/former)** | 19/1156 (1.6%) |
| **Daily alcohol use** | 20/1147 (1.7%) |

Data are described as: n (%), n/N (%), mean (±SD), or median (IQR).

\*Some patients were diagnosed using more than one test.

\*\* Chronic Kidney Disease, Neoplasia, HIV, neurological disease.

corresponding to a Medical Research Council Dyspnea score ≥2. Other symptoms frequently reported were fatigue (63.4%), memory loss (54.5%), insomnia (52.7%), and chest pain (42.8%) (Table 2). Women had a higher fatigue and headache frequency across all disease severity groups (Fig 2 and Table 2). Additionally, higher proportions of pain symptoms (myalgia, headache, chest pain) were observed among women with moderate and severe acute disease than men. No differences in long COVID symptoms were observed between ethnic groups (black/mixed-race vs. white and others). Long COVID complaints are usually initiated at the acute stage of disease, persisting as residual symptoms. Some patients with long COVID reported new symptoms after recovery from acute illness, including chest pain, myalgia, and fatigue (S1 Table and S1 Fig).

Clinical presentation of Long COVID varied with calendar time. Olfactory and gustatory dysfunctions were more frequently detected during the circulation of ancestral strain compared to the period of gamma/delta variant (S2 Table).

**Table 2. Persistent symptoms separated by sex and disease severity at the acute phase.**

| | | Mild | | | Moderate | | | Severe | | |
|---|---|---|---|---|---|---|---|---|---|---|
| | OVERALL | Male (N = 89) | Female (N = 262) | Adj. p | Male (N = 155) | Female (N = 183) | Adj, p | Male (N = 262) | Female (N = 213) | Adj. p |
| Time of disease onset (months) | 2.3 (1.6–3.7) | 2.9 (1.6–4.4) | 2.6 (1.6–4.8) | 0.827 | 2.0 (1.5–3.4) | 2.0 (1.5–3.5) | 0.507 | 2.3 (1.7–3.2) | 2.5 (1.8–3.6) | 0.100 |
| Time until hospitalization (days) | 9 (5–15) | — | — | — | 10.0 (5.0–16.8) | 10.0 (5.0–17.0) | 0.800 | 9.0 (5.0–15.0) | 8.0 (5.0–13.2) | 0.440 |
| Length of stay in hospital (days) | 11 (6–18) | — | — | — | 7.0 (5.0–10.0) | 6.0 (4.8–10.0) | 0.295 | 15.0 (10.0–23.0) | 15.0 (10.0–25.5) | 0.558 |
| Age, years | 52.1 (±13.4) | 48.0 (±13.0) | 47.4 (±11.6) | 0.769 | 55.0 (±14.0) | 54.0 (±14.1) | 0.597 | 54.0 (±12.6) | 53.7 (±13.9) | 0.855 |
| BMI- kg/m$^2$ | 29.9 (±6.1) | 28.1 (±4.3) | 29.3 (±6.1) | 0.151 | 28.4 (±5.8) | 31.4 (±7.0) | **<0.001** | 29.2 (±5.0) | 31.7 (±6.4) | **<0.001** |
| Any comorbidity | 844 (72.6) | 48 (53.9) | 170 (64.9) | 0.117 | 107 (69.0) | 149 (81.4) | **0.015** | 195 (74.4) | 176 (82.6) | 0.067 |
| Number of persistent symptoms | 4.0 (2.0–6.0) | 3.0 (2.0–5.0) | 5.0 (3.0–7.0) | **0.008** | 3.0 (1.0–5.0) | 5.0 (3.0–7.0) | **<0.001** | 3.0 (2.0–5.0) | 5.0 (3.0–7.0) | **<0.001** |
| Headache | 411/1112 (37) | 32 (36.0) | 131/260 (50.4) | 0.062 | 36/151 (23.8) | 77/175 (44.0) | **0.001** | 58/236 (24.6) | 76/201 (37.8) | **0.011** |
| Cough | 453/1163 (39) | 35 (39.3) | 97 (37.0) | 0.803 | 56 (36.1) | 75 (41.0) | 0.451 | 101/261 (38.7) | 89 (41.8) | 0.538 |
| Dyspnea | 790 (67.9) | 58 (65.2) | 178 (67.9) | 0.763 | 98 (63.2) | 128 (69.9) | 0.281 | 188 (71.8) | 139 (65.3) | 0.213 |
| mMRC≥2[1] | 361/744 (48.5) | 22 (37.9) | 75 (44.6) | 0.505 | 36 (38.3) | 56 (46.7) | 0.306 | 93/172 (54.1) | 78/131 (59.5) | 0.426 |
| Myalgia (excluding Chest pain) | 457/1163 (39.3) | 26 (29.2) | 106 (40.5) | 0.121 | 44 (28.4) | 96 (52.5) | **<0.001** | 85/261 (32.6) | 100 (46.9) | **0.008** |
| Fatigue | 738/1163 (63.5) | 53 (59.6) | 194 (74.0) | 0.038 | 77 (49.7) | 121 (66.1) | **0.006** | 143/261 (54.8) | 150 (70.4) | **0.006** |
| Chest pain | 525/1162 (45.2) | 53 (59.6) | 149 (56.9) | 0.117 | 54 (34.8) | 88/182 (48.4) | 0.021 | 84/261 (32.2) | 97 (45.5) | **0.011** |
| Dysphagia | 55/1143 (4.8) | 4/88 (4.5) | 17/259 (6.6) | 0.781 | 7/153 (4.6) | 7/182 (3.8) | 0.789 | 4/255 (1.6) | 16/206 (7.8) | **0.008** |
| Dysphonia | 61/1143 (5.3) | 3/88 (3.4) | 13/257 (5.1) | 0.805 | 6/153 (3.9) | 6/182 (3.3) | 0.808 | 15/257 (5.8) | 19/206 (9.2) | 0.290 |
| Gustatory disfunction | 167/1029 (16.2) | 9/84 (10.7) | 73/249 (29.3) | **0.005** | 8/138 (5.8) | 28/158 (17.7) | **0.006** | 24/221 (10.9) | 25/179 (14.0) | 0.412 |
| Olfactory disfunction | 174/1006 (17.3) | 13/82 (15.9) | 76/245 (31.0) | **0.037** | 10/131 (7.6) | 32/154 (20.8) | **0.006** | 20/219 (9.1) | 23/175 (13.1) | 0.301 |
| Motor disabilities | 193/1028 (18.8) | 5/83 (6.0) | 37/248 (14.9) | 0.092 | 14/138 (10.1) | 37/158 (23.4) | **0.006** | 51/222 (23.0) | 50/179 (27.9) | 0.392 |
| Loss of appetite | 176/1148 (15.3) | 7 (7.9) | 53/259 (20.5) | **0.040** | 13/153 (8.5) | 31/182 (17.0) | **0.035** | 34/258 (13.2) | 37/207 (17.9) | 0.253 |
| Hair loss | 230/594 (38.7) | 2/37 (5.4) | 44/95 (46.3) | **<0.001** | 12/83 (14.5) | 53/93 (57) | **<0.001** | 28/151 (18.5) | 90/134 (67.2) | **<0.001** |
| Dizziness | 212/602 (35.2) | 12/37 (32.4) | 44/97 (45.4) | 0.252 | 15/84 (17.9) | 36/93 (38.7) | **0.006** | 44/155 (28.4) | 61/136 (44.9) | **0.011** |
| Insomnia | 317/603 (52.6) | 18/37 (48.6) | 62/98 (63.3) | 0.189 | 32/84 (38.1) | 60/93 (64.5) | **0.002** | 70/155 (45.2) | 76/136 (55.9) | 0.121 |
| Memory loss | 332/603 (55.1) | 16/37 (43.2) | 60/98 (61.2) | 0.115 | 38/85 (44.7) | 66/93 (71) | **0.002** | 68/154 (44.2) | 83/136 (61) | **0.015** |
| Oxygen Saturation (Pulse Oximetry) | 97 (96–98) | 97.0 (97.0–98.0) | 98.0 (97.0–99.0) | 0.089 | 97.0 (96.0–98.0) | 97.0 (96.0–98.0) | 0.291 | 97.0 (96.0–98.0) | 97.0 (96.5–98.0) | **0.066** |

Data are described in n(%), n/N(%), mean (±sd), or median (IQR). P-values were adjusted for multiple comparisons with Benjamini–Hochberg procedure.

1- Only applied to patients reporting dyspnea.

Due to technical problems, HbA1c measurement was performed for only 664 (57.0%) patients. A total of 426 (64.1%) exhibited altered levels (above 5.7%) of HbA1c, a median of 2.4 (1.7–4.2) months after disease onset (Table 3). High levels (above 6.4%) were detected for 175

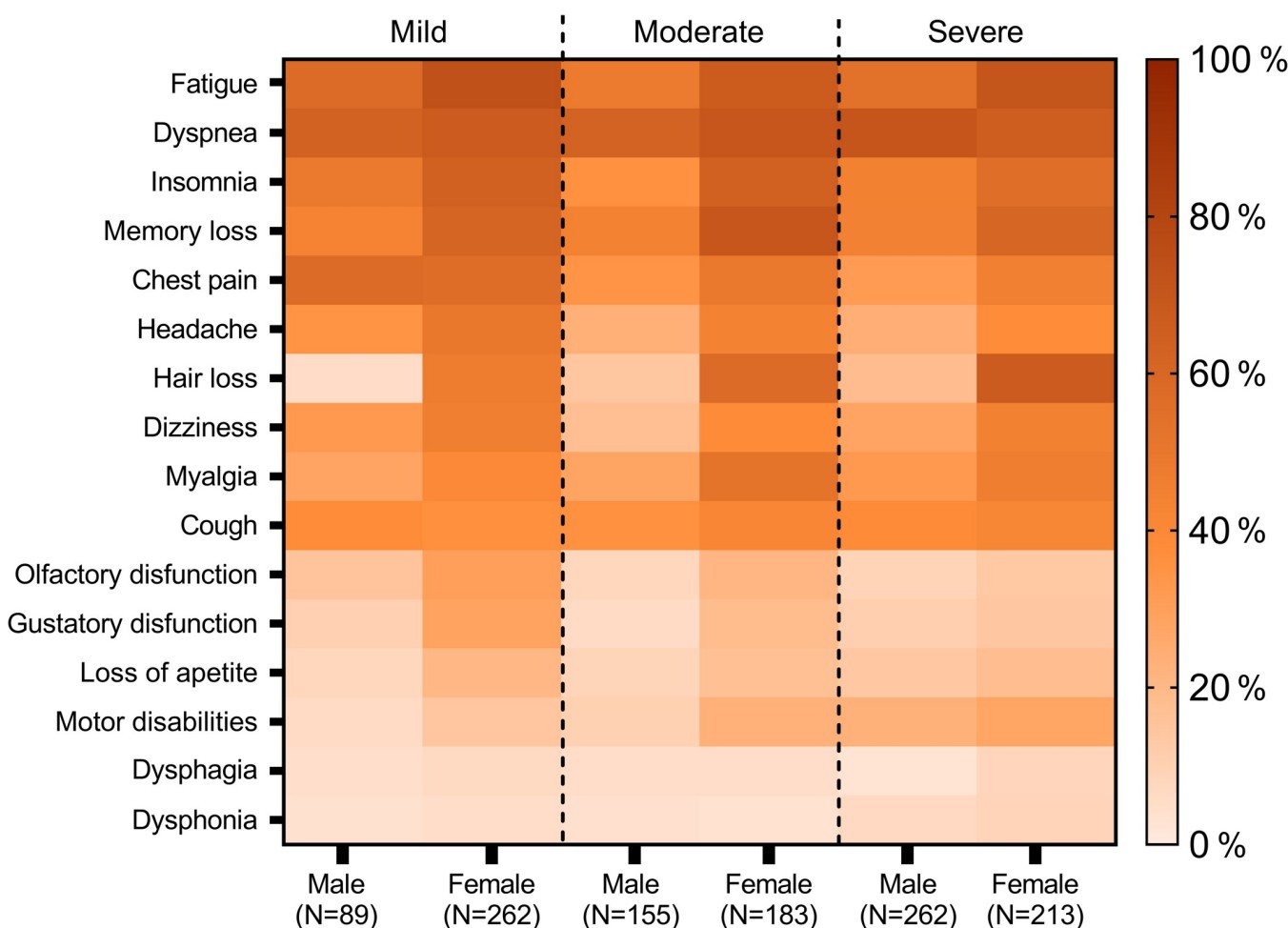

**Fig 2. Heat map illustrating the frequency of residual symptoms among patients long COVID patients according to sex and disease severity.**

patients; 71 (40%) of them denied a previous diagnosis of diabetes mellitus (DM). HbA1c levels did not correlate with time after disease onset (S2 Fig). Use of corticosteroids, anemia, and obesity/comorbidities were similar among long COVID cases with and without high levels of HbA1c (S3 Table). Other common biochemical alterations included increased levels of C-reactive protein [324/760 (42.6%)] and low levels of Hb [248/896 (27.7%)] at the time of the first appointment at CPC. CRP, hemoglobin, and HbA1c remain the most frequent laboratory alteration regardless of time since symptom onset (S4 Table).

A total of 51 (4.0%) patients with long COVID symptoms required post-acute hospitalization, the majority (n = 38, 74.5%) between 1 to 3 months after disease onset, mainly due to dyspnea/chest pain (n = 17), infection (n = 13) and pulmonary thromboembolism (n = 3) (S5 Table). Among them, eight patients presented mild acute disease.

The influence of long COVID on patients' quality of life was evaluated by EuroQoL (EQ-5D-5L) in 929/1164 patients with different classifications of COVID-19 disease severity. Of these, 826 (88.9%) reported some degree of alteration, 98 (8.4%) with inabilities or extreme values in at least one domain of the EQ-5D-5L. The most affected dimensions were anxiety/depression [207/924 (22.4%) presenting severe or extreme anxiety] and pain/discomfort [135/924 (17.6%) severe or extreme pain] (S6 Table). The median EuroQoL Global Score was 70

**Table 3. Laboratory findings of blood samples collected during first visit to CPC.**

| | Total n/N (%) | ACUTE DISEASE SEVERITY | | |
| --- | --- | --- | --- | --- |
| | | **Mild** | **Moderate** | **Severe** |
| | | **n/N (%)** | **n/N (%)** | **n/N (%)** |
| **Hemoglobin** (<14/12 g/dL)* | 248/896 (27.7) | 41/291 (14.1) | 73/263 (27.8) | 134/342 (39.2) |
| **White Blood Cell count per μl** (> 3500 or < 10500) | 98/895 (10.9) | 22/291 (7.6) | 30/262 (11.5) | 46/342 (13.5) |
| **Lymphocytes count per μl** (>700 or <4410) | 70/875 (8.0) | 28/289 (9.7) | 22/256 (8.6) | 20/330 (6.1) |
| **Platelet** (<140.000/dl or >450.000/dl) | 25/894 (2.8) | 5/289 (1.7) | 4/263 (1.5) | 16/342 (4.7) |
| **Urea** (<19/15 mg/dL or >43/36 mg/dL)* | 164/873 (18.4) | 41/278 (14.7) | 48/259 (18.5) | 75/336 (22.3) |
| **Creatinine** (>0.8/0.7mg/dL or <1.5/1.2 mg/dL)* | 161/877 (18.8) | 54/283 (19.1) | 43/256 (16.8) | 64/338 (18.9) |
| **Sodium** (<135 or >150 mEq/L) | 34/864 (3.9) | 16/277 (5.8) | 11/258 (4.3) | 7/329 (2.1) |
| **Potassium** (<3.5 or >5.0 mEq/L) | 86/862 (10.0) | 21/280 (7.5) | 27/255 (10.6) | 38/327 (11.6) |
| **Creatinine Phosphokinase** (>170/135 U/L)* | 107/835 (12.8) | 53/274 (19.3) | 23/244 (9.4) | 31/317 (9.8) |
| **C-Reactive protein** (>5mg/L) | 324/760 (42.6) | 82/247 (33.2) | 108/224 (48.2) | 134/289 (46.4) |
| **Alanine Transaminase** (>72/52 U/L)* | 60/869 (6.9) | 22/287 (7.7) | 18/255 (7.1) | 20/327 (6.1) |
| **Aspartate Transaminase** (>59/36 U/L)* | 41/872 (4.7) | 10/289 (3.5) | 15/254 (5.9) | 16/329 (4.9) |
| **Albumin** (>3.2 g/dL or >4.8 g/dL) | 36/697 (5.2) | 14/238 (5.9) | 8/205 (3.9) | 14/254 (5.5) |
| **Total bilirubin** (>1.2 mg/dL) | 25/766 (3.3) | 9/255 (3.5) | 7/225 (3.1) | 9/286 (3.1) |
| **Glycated Hemoglobin** | N = 664 | N = 219 | N = 196 | N = 249 |
| . <5.7% | 238 (35.8) | 118 (53.9) | 52 (26.5) | 68 (27.3) |
| . 5.7–6.4% | 251 (37.8) | 68 (31.1) | 81 (41.3) | 102 (41.0) |
| . >6.4% | 175 (26.4) | 33 (15.1) | 63 (32.1) | 79 (31.7) |

Data are described in n/N(%). n = altered laboratory exam based on laboratory reference values.

* Reference values for male and female, respectively.

[IQR 50–80]. We further evaluated associations between clinical and demographic characteristics concerning the impact on health-related quality of life. The female sex was associated with worse scores across all domains. As expected, the domains of mobility and self-care were more affected in patients with severe acute disease. The presence of comorbidities influenced only the domain of pain/discomfort. The most prevalent persistent symptoms (dyspnea, chest pain, and fatigue) were associated with decreased quality of life. In individuals reporting fatigue, all five domains of the EuroQol were negatively affected (Fig 3, S7, and S8 Tables). Fatigue seems to have a major impact on QoL since self-care, usual abilities, and pain EuroQoL domains were more affected in patients reporting this symptom.

## Discussion

To the best of our knowledge, the present report represents the first large-scale characterization of long COVID in a sample of Latin American patients, mainly mixed-race or black. Our analysis showed that long COVID mainly affected women, causing dyspnea, fatigue, and chest pain regardless of COVID-19 severity, but no change was observed by race and ethnicity. By contrast, the prevalence of long COVID has been significantly lower in mild cases [20]. However, our findings are consistent with reports on patients from North America, Asia, and Europe [21, 22]. Interestingly, glucose metabolism disorder increased inflammatory markers, and anemia was frequently observed at the time of admission at post COVID center. Some patients were readmitted to hospitals later after the disease onset.

Among the biochemical abnormalities detected in this population with long COVID, elevated levels of HbA1c were a frequent finding in individuals without a previous diagnosis of

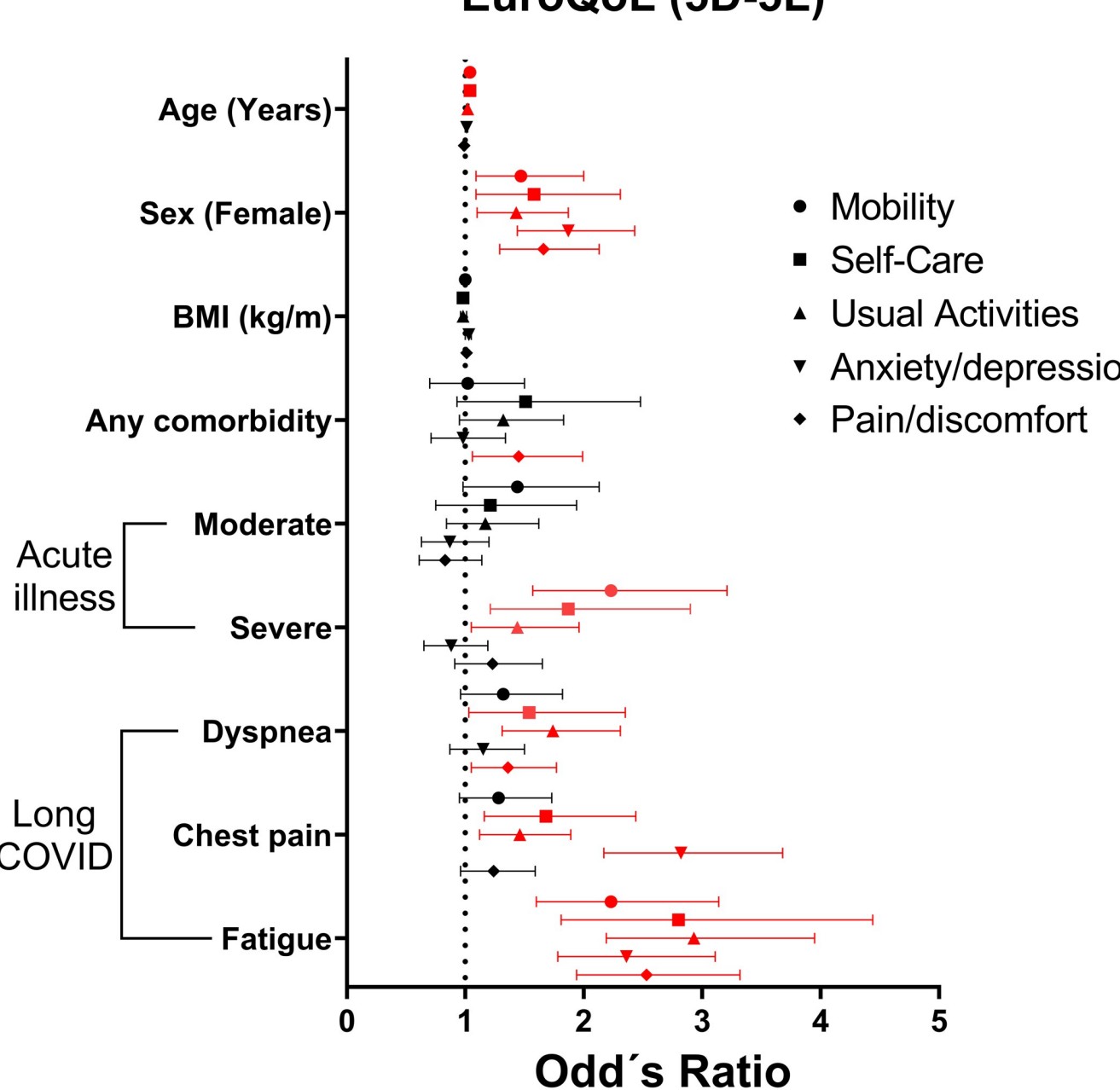

**Fig 3. Forest plot of ordinal logistic regression for each domain of EuroQoL (Mobily, Self-care, Usual Activities, Anxiety/Depression, and Pain/discomfort).** Each symbol represents the odds ratio with the corresponding 95% confidence interval of presenting alteration (≥2 points) in each domain of EuroQoL separated by predictor variables. Significant results (p<0.05) are highlighted with a red line. Descriptive results are displayed in the supplementary material.

DM. The onset of DM after COVID-19 has been postulated. SATHISH *et al*. [23] reported that 14.4% of 3,711 patients hospitalized due to COVID-19 had been recently-diagnosed with DM. Huang *et al*. [7] reported 58 out of 1526 patients presenting newly diagnosed DM after discharge, with one case presenting the normal level of HbA1c during the acute phase and altered levels at follow-up. Possible causes may include increased insulin resistance, elevated ACE2

levels, and inflammatory and autoimmune conditions. However, some confounding factors must be taken into account. HbA1c may appear to be misleadingly increased due to the lifespan of red blood cells (such as in anemia related to iron deficiency) [24]. Glucocorticoid use should also be considered [23]. Indeed, anemia was the most frequent laboratory alteration in our sample (27.7%), and corticosteroid use was reported by 72.4% of our patients. However, we did not observe a difference in the frequency of confounding factors among patients with and without altered HbA1c.

Additionally, higher HbA1c levels are expected in specific ethnic populations, such as blacks and Latin Americans [25]. Since data on previous HbA1c levels (prior to COVID-19 or during acute infection) was unavailable, previously undiagnosed DM must be contemplated as an explanation. Furthermore, further studies should assess whether Brazilian patients chronically experience difficulties in accessing public health care services. Finally, our finding of frequently altered levels of HbA1c reinforces the importance of screening for metabolic disorders in cases of long COVID.

Some patients with long COVID required post-acute hospitalization due to respiratory infection, chest pain, dyspnea, thrombosis, among others. A large UK cohort found COVID-19 increased the risk of rehospitalization in patients post-discharge in comparison to matched controls from the general population [(aHR) 2.22, 2.14 to 2.30] [26]. Hospitalization after discharge in patients with moderate-to-severe infection may be related to higher frequencies of comorbidities in this population [27]. A recent meta-analysis reported a 9.3% readmission rate (95%CI: 5.5%-15.4%) associated with risk factors identified as male sex, older age, and previous comorbidity, including COPD [28]. Indeed, among the 23 patients with COPD studied herein, five (22%) were hospitalized about one month after disease onset, suggesting that this clinical condition may represent a risk factor for further complications. Moreover, among the eight individuals with mild acute disease who had post-acute hospitalization, comorbidities were detected in seven: four presented obesity, three hypertension, and one DM. Together these data suggest that long COVID associated with comorbidities may place patients at increased risk of hospitalization and require close monitoring. However, the data needed for postulating that long COVID exposes patients to a greater risk of post-acute hospitalization are still necessary.

Furthermore, considering that early hospital readmissions have been used as solid indicators of a lower quality of hospital care and follow-up [29], we found that 38 (3%) patients were readmitted early after acute infection, possibly indicating premature hospital discharge. Together, the detection of metabolic disorders and high post-acute hospitalization rates may reflect longstanding deficiencies in the Brazilian public health care system, aggravated by the demands imposed by the COVID-19 pandemic. Thus, more severe consequences of long COVID and its impact on quality of life should be expected in low-income countries when associated with poor management of comorbidities and insufficient access to health care services.

Patients from low/medium income countries of black/mixed-race have been sporadically represented in descriptive studies of long COVID. Two small studies involving Brazilian patients (each with less than 201 patients) revealed symptomatology similar to that described herein [12, 13], as well as comparable ethnic characteristics and impact on QoL [12, 13]. Thus, our findings validate previously published evidence in a large-size sample of mostly black and mixed-race individuals. Moreover, our data demonstrate that long COVID may exacerbate chronic health care system deficiencies. The Brazilian population presents a high prevalence of metabolic disorders, poor control of comorbidities [30], and higher rates of undiagnosed DM cases than developed countries [31], possibly due to a lack of outpatient screening and follow-up.

Regardless of disease severity, women presented more residual symptoms as evidenced by higher frequencies of fatigue, chest pain, and myalgia, which agree with previously published data [7, 29]. Women also reported worse quality of life scores across all domains of the Euro-QoL (mobility, self-care, usual activities, anxiety/depression, and pain). A cohort study found that anxiety and depression were also associated with persistent fatigue in females up to six months after COVID-19 disease onset [7]. A Chinese study focusing on QoL in COVID patients found the female sex to be an independent predictor of worse quality of life [OR: 1.79, 95%CI: 1.04–3.06] [32]. In sum, these factors may exert bidirectional effects on long COVID symptomatology since psychological disorders are thought to exacerbate residual symptoms [33]. In addition, social studies have shown that women are more likely to report symptoms and seek health services than men, which may be a potential confounder in the preponderance of females suffering from long COVID [34]. Autoimmunity may also be implicated, as suggested by identifying higher titers of antinuclear antibodies in women with persistent symptoms [35]. However, further research is required to more comprehensively understand the role played by autoimmunity in the prevalence of females with long COVID.

Reports of dyspnea are frequent in long COVID patients, mainly after acute respiratory distress syndrome [8]. Our data indicate residual breathlessness despite normal resting oxygen saturation and regardless of disease severity during the acute stage. Post-acute dyspnea apparently has a multifactorial etiology. In addition to lung disorders with abnormalities in chest computed tomography and lung function, anxiety, hyperventilation syndrome, respiratory muscle weakness, and dysfunctional breathing have also been implicated [5]. The perception of breathlessness may be confused with or potentialized by other complaints. Fatigue was reported in 70.4%% of our patients with dyspnea—a common finding in long COVID reported by 53 to 73% of SARS-CoV-2-affected patients [7, 8]. Immunological dysregulation is implicated in both dyspnea and fatigue. Chronic infection and iatrogenic complications following severe disease (e.g., long periods of immobility and corticosteroid use) are recognized risk factors for muscle wasting and dysfunction [36, 37]. Long-term post-COVID cardiovascular complications have also been postulated and may be associated with cardiorespiratory complaints, including in non-hospitalized patients [38]. Our analysis indicated that dyspnea and chest pain were related to reductions in quality of life; however, fatigue, reported in 50 to 74%, was associated with the lowest EuroQol Global scores. It has been proposed that pain symptoms may be related to neuroinflammation, muscle lesion, or direct viral injury. Longstanding pain may require hospitalization for treatment and could lead to severe conditions that negatively impact patient health and QoL.

Our study has several limitations. First, we did not investigate sleep disorders and other neurological sequelae that have been associated with long COVID. The CPC resides in a reference hospital for respiratory diseases, and cases with respiratory complaints may have been preferentially referred to this health unit. However, it´s important to highlight that the CPC was the reference unit for long COVID in the Bahia State, which may have attenuated the selection bias. Indeed, several non-respiratory post-COVID symptoms were detected in our study, as in other reports from the literature. Second, our current study was unable to assess the prevalence of COVID in the population, nor the differences in clinical presentation of long COVID across groups of mild, moderate, and severe acute infection. This limitation is related to the strategy of recruitment. The mild acute phase patients (outpatients) were self-referred/referred in, probably selecting cases with more severe residual symptoms and/or with more debilitating conditions. Indeed, 53% of patients with mild disease reported at least one comorbidity. Third, we were also unable to conclude if the abnormal laboratory exams/ disturbance in QoL were a direct effect of COVID-19 or due to previous conditions/stress related to the COVID-19 pandemic. Further cohort studies, including baseline exams/EuroQoL measures,

should be performed to evaluate the causality. Finally, due to limited discharge summary data, we cannot conclude if post-acute hospitalization stems from long COVID symptomatology.

## Conclusions

In conclusion, the clinical symptoms of long COVID in black and mixed-race patients were similar to previous descriptions in white and Asian populations, with similar lingering and debilitating symptoms across the spectrum of disease severity. However, the unexpectedly high frequency of post-acute hospitalization and frequent detection of glucose metabolism disorder likely resulted from the combination of long COVID and historical deficiencies in the Brazilian public health care system. Social conditions may also negatively affect QoL. Morbidity due to pain and physical/respiratory symptoms negatively affects the quality of life, mainly in women who seem more affected regardless of COVID-19 disease severity. Considering the impact of long COVID on daily activities and the high incidence of COVID-19 in Brazil, further studies should be performed to evaluate better the magnitude of this condition and its long-term impacts on the Brazilian public health system.

## Supporting information

**S1 Fig. Percentage of new symptoms reported after recovery from acute illness (late-onset symptoms) according to the disease severity.**
(TIF)

**S2 Fig. Scatter plot of HbA1C results per time since acute disease (months).** Patients with no previous diagnoses of DM were categorized according to test levels as ≤6.4% and >6.4%. Red squares are HbA1C levels of patients with previous diagnoses of DM, blue circles, no previous diagnoses of DM—HbA1C≤6.4%, and black triangles, no previous diagnoses of DM—HbA1C>6.4%. Red, blue, and black lines represent linear regression; significance levels are displayed in the graph legend.
(TIF)

**S1 Table. Acute phase symptoms of long covid patients.** Data are n (%), n/N (%), mean (±SD) or median (IQR).
(PDF)

**S2 Table. Clinical presentation by calendar time according to variant predominance distribution.** Data are described as n(%), n/N (%), or mead (±SD). Variant Period: 1- August 2020 to January 2021. 2- March 2021 to July 2021 [1].
(PDF)

**S3 Table. Clinical characteristics of patients with and without previous diagnosis of DM.** Characteristics were chosen according to the potential to modulate HbA1C level. Group of patients were divided according to the level of HA1c. Data are in n (%) or n/N (%), in case of missing data. Only patients that presented hbA1C results were described. P-value <0.05 are highlighted. 1- "Corticosteroid use" refers to specific use at acute phase of disease. 2- Diagnostic based on the level of hemoglobin measured at first visit. 3- Full list of comorbidities is described in Table 1. 4- Newly diagnoses of DM.
(PDF)

**S4 Table. Laboratory blood analysis divided by time since acute disease.** Data are in n/N (%). All reference values used to define laboratorial alteration are described in parenthesis. *Reference values for male and female, respectively. **N = 664.
(PDF)

**S5 Table. Characteristics of patients hospitalized one month or later after disease onset.** Data are n (%) or mean (±SD). (PDF)

**S6 Table. Descriptive EuroQol results separated by sex and disease severity at the acute phase.** Data are n (%) or mean (±SD). "N" represents the total sample of the subgroup. 1- P-value of analysis of each EuroQoL domains levels compared by sex and disease severity. (PDF)

**S7 Table. Ordinal logistic regression for EuroQoL domains.** (PDF)

**S8 Table. Multiple linear regression for EuroQoL global score.** (PDF)

**S1 Dataset. Study's minimal underlying data set.** (XLSX)

# Acknowledgments

The authors are grateful to those involved in the patient's clinical care, the nursing, administrative technicians, and all members of the CPC study group. We thank all patients who participated in this study and their families. The authors also thank Andris K. Walter for English language revision and manuscript copy editing assistance.

**CPC study group**:

Maria de Fátima Estrela[1], Mariângela Ramos[1], Cristiane Mesquita[1], Marizete Matos[1], Manoela Fontes[1], Juliane Serra[1], Maristela Sestelo[1], Marcia Marinho[1], Patrícia Santos[1], Carolina Neves[1], Eliana Matos[1], Fabiane Fontoura[1], Thaiana Ramos[1], Yasmin Cadidé[1], Anna Diniz[1], Kristine Mendes[1], Juliana Sant'ana[1], Andréa Jesus[1], Marta Santos[1], Carla Guedes[1], Joelma Anunciação[1], Vânia Pedreira, Margaranei Reis[1], Monica Gomes[1], Emília Passos[1], Priscila Bahia[1], Milena Azevedo[1], Giancarla Credico[1], Adriele Bastos[1], Eliene Gibaut[1], Alisson Andrade[1], Antônio Silva[1], Maria de Fátima Silva[1], Denise Albuquerque[1], Estela Cordeiro[1], Jamile Oliveira[1], Monique Oliveira[1], Valcelia Muniz[1] and Juliane Simões[1].

**Affiliation**:

1. Hospital Especializado Octávio Mangabeira (HEOM), Salvador, Brazil.

**CPC leader**:

**Ana Paula Andrade Barreto**

Address: Manoel Gomes de Mendonça street, 307, apt 1301, Pituba

Zip code: 41810–820

# Author Contributions

**Conceptualization:** Aquiles Camelier, Manoel Barral-Netto, Viviane Boaventura, Marcelo Chalhoub Coelho Lima.

**Data curation:** Ana Paula Andrade Barreto, Marcio Andrade Barreto Filho, Thiago Cerqueira-Silva.

**Formal analysis:** Thiago Cerqueira-Silva.

**Funding acquisition:** Marcelo Chalhoub Coelho Lima.

**Writing – original draft:** Ana Paula Andrade Barreto, Marcio Andrade Barreto Filho, Lucimeire Cardoso Duarte, Thiago Cerqueira-Silva, Viviane Boaventura.

**Writing – review & editing:** Aquiles Camelier, Natalia Machado Tavares, Manoel Barral-Netto, Marcelo Chalhoub Coelho Lima.

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
