## [Decision Letter · Decision Letter 0]

2 Aug 2022

PONE-D-22-10158

Metabolic disorders and re-hospitalization in black/mixed-race patients with long COVID in Brazil

PLOS ONE

Dear Dr. Boaventura

Thank you for submitting your manuscript to PLOS ONE. After careful consideration, we feel that it has merit but does not fully meet PLOS ONE’s publication criteria as it currently stands. Therefore, we invite you to submit a revised version of the manuscript that addresses the points raised during the review process.

We look forward to receiving your revised manuscript.

Kind regards,

Lucien Peroni Gualdi, PhD

Academic Editor

PLOS ONE

“MABF received a Scientific Initiation Scholarship - PIBIC from “Conselho Nacional de Desenvolvimento Científico e Tecnológico” (CNPq) -Brazil; T.C.-S. is a PhD student at the Post-Graduation Program in Health Sciences-UFBA, which is supported by the “Coordenação de Aperfeiçoamento de Pessoal de Nível Superior”-Brasil, finance code 001. A C. reports consulting fees from “Fundação Maria Emilia Freire”. MB-N and VSB are research fellows from CNPq.”

5. One of the noted authors is a group or consortium [CPC study group]. In addition to naming the author group, please list the individual authors and affiliations within this group in the acknowledgments section of your manuscript. Please also indicate clearly a lead author for this group along with a contact email address.

Additional Editor Comments:

The manuscript has been assessed by the reviewers and several questions were raised by them. The authors may answer all questions before consideration to publication in the journal. All questions raised can be found in the authors menu.

Reviewers' comments:

Reviewer's Responses to Questions

**Comments to the Author**

1. Is the manuscript technically sound, and do the data support the conclusions?

Reviewer #1: Partly

Reviewer #2: Yes

Reviewer #3: Yes

2. Has the statistical analysis been performed appropriately and rigorously? 

Reviewer #1: No

Reviewer #2: Yes

Reviewer #3: Yes

3. Have the authors made all data underlying the findings in their manuscript fully available?

Reviewer #1: No

Reviewer #2: Yes

Reviewer #3: Yes

4. Is the manuscript presented in an intelligible fashion and written in standard English?

Reviewer #1: Yes

Reviewer #2: Yes

Reviewer #3: Yes

5. Review Comments to the Author

Reviewer #1: Thank you for the opportunity to review the manuscript “Metabolic disorders and re-hospitalization in black/mixed-race patients with long COVID in Brazil”. This study describes a cohort of adults seeking care in a post COVID care clinic in Brazil between August 2020 and September 2021. Outcomes evaluated included hospital readmission, HbA1c levels, long COVID symptoms and validated quality of life survey results. The cohort included patients who were outpatients as well as inpatients for their acute COVID infection. This study provides insights into long COVID in an understudied population (LMIC, black and mixed race), with findings that suggest that there is little difference in presentation by race/ethnicity, and provides interesting laboratory data about this important population. However, there are several limitations worth noting that dampen my overall enthusiasm for this manuscript.

1. The population evaluated is a mix of inpatients, all of whom are referred to the clinic, and outpatients, who comprise individuals are seeking medical attention due to their symptoms, preventing any real conclusions to be drawn from the comparisons of mild and moderate/severe disease, or the frequency of symptoms in this population.

2. The population is biased based on the above reasons, but also may over-represent patients with respiratory complaints.

3. The absence of a control group prevents true understanding of which symptoms and QoL measures may be a direct effect of SARS CoV2 infection, versus the situational context of the pandemic.

4. The assessment of readmissions is incomplete and non-contributory- as it is not known whether these readmissions were related to long COVID symptoms or underlying conditions.

5. There was no standardized follow up period, making the laboratory value interpretation difficult, as well as the reported symptoms. Some stratifying by time period after acute infection may therefore be helpful.

6. Interpretation of HbA1c is also difficult given the summary data doesn’t take into account those with existing diabetes, nor the time after acute infection, or other factors that may influence these results (such as Hemoglobin, corticosteroid use and patient risk factors) and there are no baseline data available for comparison.

Minor comments are outlined below:

Introduction:

Please state the objectives in the final paragraph more clearly, eg. That the main objectives were to describe the symptoms associated with long COVID, and include a description of which complications were being evaluated.

Methods:

A description of which patients are being seen at CPC would be helpful in the methods, that all inpatients are referred, but only those with persistent symptoms with outpatient COVID are being self referred/referred in.

Would recommend excluding patients with clinical features of viral pneumonia, since this is not discriminative enough for COVID-19 infection.

Statistical analyses: ordinal logistic regression analyses did not account for underlying medical condition, which is another important covariate to consider.

Results:

Suspected COVID-19 included those with a CT showing viral infection, but 16 patients were excluded due to no confirmatory diagnosis of COVID-19 which is confusing and doesn’t match the inclusion/exclusion criteria from the methods.

“residual symptoms are usually initiated at the acute stage of disease”- residual by definition implies that the same symptoms were present during acute illness, so this statement is better switched to “new symptoms including chest pain, myalgia and fatigue were reported after recovery from acute illness”. However, this may be subject to recall bias.

Table 2- there are many analyses included here, increasing the likelihood of chance findings, and a test for multiple comparisons would be beneficial.

The HbA1c data would be helpful to be represented as a scatter plot, with the y axis being the HbA1c value, the x axis being time since acute infection in weeks/months, and two different sets of data including 1) those with a history of diabetes and 2) those values among patients without a history of diabetes.

Table 3. Suggest stratifying laboratory findings based on outpatient, hospitalized and ICU admission for acute COVID, which may be more interesting to readers.

“Late hospitalization” is a confusing term, and the statement about 8 patients presenting with mild COVID-19 warrants clarification. Isn’t mild defined here as outpatient illness? And re-testing within 90 days could represent viral shedding rather than new infection.

Were there any changes in presentation, symptom reporting by calendar time?

Figure 3- EuroQoL plot is difficult to follow, suggest some re-organizing with a figure legend description.

Discussion:

In the main findings paragraph, would also include that there was no change by race and ethnicity.

Discussion, paragraph 2 - Huang report – “reported 58…” – need denominator to be included in this sentence.

Page 13, line 16. Do not understand the term “chronic health system deficiencies” – does this mean “underlying medical conditions”?

Limitations- would include additional limitations outline above.

Additional comments: Several typographical errors are noted throughout, eg. Symp-toms (line 10, abstract), c-protein reactive instead of C- reactive protein (results).

Reviewer #2: Many thanks for asking me to review this manuscript entitled "Metabolic disorders and re-hospitalization in black/mixed-race patients with long COVID in Brazil "

The approach used by authors is comprehensive and could be interesting within a process of quality improvement in the management of the post-COVID resulting from mild to severe COVID-19 which is why I enjoyed reading this manuscript.

1/ Title is clear but should display the study’s design.

2/ Abstract.

Authors provided along with an informative and balanced summary of what was done and what was found.

Authors may just reorganize abstract in a structured manner.

Second sentence is an only very long methods-related phrase !

The aim of the study should be individualized.

Results should be presented in a uniform manner, n(%) (The example of HbA1c with no %).

3/ Introduction

Authors explained well the scientific background and rationale for the investigation being reported and stated specific objectives.

4/ Methods

Authors presented key elements of study design early in the section. They described the setting, locations, and relevant dates, including periods of recruitment and data collection. They gave the eligibility criteria, and the sources and methods of selection of participants. They clearly addressed needed definitions. They explained how the study size was arrived at. They described all statistical methods, including those used to control for confounding.

Authors are encouraged to describe more their efforts to address potential sources of bias and explain how missing data were addressed in this section.

5/ Results

The flow diagram is appreciated

Table 2, I suggest authors adding a column with non separated values.

All results are given and displayed within comprehensive and systemic tables and figures. Patients characteristics, information on exposures and potential confounders are given.

They indicated number of participants with missing data for each variable of interest.

6/ Discussion

Authors summarized key results with reference to study objectives.

Authors addressed cautious overall interpretation of results considering objectives, limitations, multiplicity of analyses, results from similar studies, and other relevant evidence according to Brazil population.

Authors are suggested to start with dyspnea, then glucose intolerance, then QoL as the order adopted in the abstract and results sections ; unless they think that glucose intolerance is more important to start with.

Limitations were well discussed.

Reviewer #3: Ana Paula Andrade Barreto etal did a cross-sectional analysis with a detailed clinical picture for people with long COVID or post-COVID-19 condition (WHO definition). A total of 1164 (88% black/mixed-race) were evaluated 2.3 [IQR=1.6-3.7] months after mild (30.2%), moderate (29.0%) or severe (40.8%) acute illness. They concluded that in this majority-black/mixed-race population in Brazil, long COVID was associated with re-hospitalization, newly diagnosed diabetes mellitus, and decreased quality of life, particularly in women and regardless of disease severity of acute infection. These findings with rigorous statistical analysis in a large-size sample validate previous published evidence of mostly black and mixed-race individuals with long COVID. Given that COVID-19 now affects millions of people worldwide, long COVID has become a meaningful public health concern and may exacerbate chronic health system deficiencies in low-middle income countries. Investigation of long COVID with valuable clinical manifestation provides important implications for health care system and development of effective treatment for this complicated syndrome.

Suggestion and Question:

1, Some longitudinal analysis on recovery from long COVID in this group of people will make this manuscript more valuable and informative, but it is not necessary for publication.

2, In Supplementary Table 3, please clarify the P value following N number for which group of data.

6. PLOS authors have the option to publish the peer review history of their article (what does this mean?). If published, this will include your full peer review and any attached files.

Reviewer #1: No

Reviewer #2: **Yes: **Mohamed Boussarsar

Reviewer #3: No

---

## [Author Response · Author response to Decision Letter 0]

6 Sep 2022

Dear Lucien Peroni Gualdi,

Academic Editor

PLOS ONE

Thanks for sending us the reviewers’ comments on our manuscript “Metabolic disorders and re-hospitalization in black/mixed-race patients with long COVID in Brazil”. We are pleased that the referees and the editors found our work interesting and are grateful for their constructive comments for improving the paper. 

We have carefully revised the manuscript, and a detailed point-by-point response to the editor and reviewers is provided in the updated document labeled "Response to Reviewers", with the changes in the manuscript text highlighted in a separate document. 

We look forward to hearing a positive response to this manuscript.

Best regards, 

Viviane Sampaio Boaventura

---

## [Editor Report · Decision Letter 1]

14 Oct 2022

Metabolic disorders and post-acute hospitalization in black/mixed-race patients with long COVID in Brazil: a cross-sectional analysis

PONE-D-22-10158R1

Dear Dr. Boaventura

We’re pleased to inform you that your manuscript has been judged scientifically suitable for publication and will be formally accepted for publication once it meets all outstanding technical requirements.

Kind regards,

Lucien Peroni Gualdi, PhD

Academic Editor

PLOS ONE

Additional Editor Comments (optional):

Please make sure your manuscript meet all the journal criteria.
---

## [Editor Report · Acceptance letter]

21 Oct 2022

PONE-D-22-10158R1 

Metabolic disorders and post-acute hospitalization in black/mixed-race patients with long COVID in Brazil: a cross-sectional analysis 

Dear Dr. Boaventura:

I'm pleased to inform you that your manuscript has been deemed suitable for publication in PLOS ONE. Congratulations! Your manuscript is now with our production department. 

Kind regards, 

on behalf of

Professor Lucien Peroni Gualdi 

Academic Editor

PLOS ONE